# SmartPill™ Administration to Assess Gastrointestinal Function after Spinal Cord Injury in a Porcine Model—A Preliminary Study

**DOI:** 10.3390/biomedicines11061660

**Published:** 2023-06-07

**Authors:** Chase A. Knibbe, Rakib Uddin Ahmed, Felicia Wilkins, Mayur Sharma, Jay Ethridge, Monique Morgan, Destiny Gibson, Kimberly B. Cooper, Dena R. Howland, Manicka V. Vadhanam, Shirish S. Barve, Steven Davison, Leslie C. Sherwood, Jack Semler, Thomas Abell, Maxwell Boakye

**Affiliations:** 1Department of Neurological Surgery, Kentucky Spinal Cord Injury Research Center, University of Louisville, Louisville, KY 40202, USA; rakibuddin.ahmed@louisville.edu (R.U.A.); fwilkins@metrohealth.org (F.W.); sharm983@umn.edu (M.S.); jay@icord.org (J.E.); monique.morgan@emory.edu (M.M.); destiny.gibson@louisville.edu (D.G.); brookecooper1@hotmail.com (K.B.C.); dena.howland@louisville.edu (D.R.H.); maxwell.boakye@uoflhealth.org (M.B.); 2Research Service, Robley Rex Veterans Affairs Medical Center, Louisville, KY 40206, USA; 3Division of Gastroenterology, Hepatology and Nutrition, Department of Internal Medicine, University of Louisville, Louisville, KY 40202, USA; manicka.vadhanam@louisville.edu (M.V.V.); shirish.barve@louisville.edu (S.S.B.); thomas.abell@louisville.edu (T.A.); 4Comparative Medicine Research Unit, University of Louisville, Louisville, KY 40202, USA; steven.davison@louisville.edu (S.D.); leslie.sherwood@louisville.edu (L.C.S.); 5Medtronic Inc., Minneapolis, MN 55432, USA; jackrsemler@outlook.com

**Keywords:** spinal cord injury, gastrointestinal motility, SmartPill™, wireless motility capsule, porcine model

## Abstract

Gastrointestinal (GI) complications, including motility disorders, metabolic deficiencies, and changes in gut microbiota following spinal cord injury (SCI), are associated with poor outcomes. After SCI, the autonomic nervous system becomes unbalanced below the level of injury and can lead to severe GI dysfunction. The SmartPill™ is a non-invasive capsule that, when ingested, transmits pH, temperature, and pressure readings that can be used to assess effects in GI function post-injury. Our minipig model allows us to assess these post-injury changes to optimize interventions and ultimately improve GI function. The aim of this study was to compare pre-injury to post-injury transit times, pH, and pressures in sections of GI tract by utilizing the SmartPill™ in three pigs after SCI at 2 and 6 weeks. Tributyrin was administered to two pigs to assess the influences on their gut microenvironment. We observed prolonged GET (Gastric Emptying Time) and CTT (Colon Transit Time), decreases in contraction frequencies (Con freq) in the antrum of the stomach, colon, and decreases in duodenal pressures post-injury. We noted increases in Sum amp generated at 2 weeks post-injury in the colon, with corresponding decreases in Con freq. We found transient changes in pH in the colon and small intestine at 2 weeks post-injury, with minimal effect on stomach pH post-injury. Prolonged GETs and CTTs can influence the absorptive profile in the gut and contribute to pathology development. This is the first pilot study to administer the SmartPill™ in minipigs in the context of SCI. Further investigations will elucidate these trends and characterize post-SCI GI function.

## 1. Introduction

SCI is associated with significant gastrointestinal (GI), urologic, dermatologic, cardiac, pulmonary, and kinematic dysfunctions that contribute to poor outcomes and often require costly medical intervention [1,2,3,4,5,6,7,8]. Mid-thoracic SCIs are associated with neurologic and GI deficits below the level of injury and can result in chronic dysfunction that frequently becomes acutely pathologic [5,9,10,11]. The impaired motility and changes in microenvironment can contribute to the deterioration of GI function, with treatment largely limited to symptom management [5,12,13,14,15,16,17,18,19,20,21].

Scintigraphy studies that utilize radioisotopes, anorectal, and colon manometry are commonly used to assess motility; however, scintigraphy is rarely used pre-clinically due to its high cost [22,23]. Substances such as charcoal or dyes can be fed to subjects and exit times noted to assess GI transit times, but are quantitatively limited to time [24,25]. Other investigations of GI function include electrogastrography and carbohydrate breath tests [26,27]. The SmartPill™ (Medtronic Inc., Minneapolis, MN, USA) is a noninvasive method for measuring gastrointestinal motility using a wireless motility capsule (WMC) that is ingested [28,29,30,31,32,33]. The WMC is a relatively new modality FDA approved for clinical assessment of gastroparesis, constipation, and general motility [34]. Pre-clinically, the WMC is used to assess pathological GI conditions and pharmacological absorption parameters [31,34,35,36,37,38]. Few studies have used the SmartPill^™^ in the context of SCI with humans or animal models [39].

The porcine spinal cord is similar to that of humans, with comparable gray to white matter ratios, cortical structure, sacral enlargement, cord dimensions/structure/organization, and metabolic demands [40,41,42,43,44,45]. This porcine SCI model serves to assess critical GI parameters, a major comorbidity in SCI patients, compare to human functionality, and consider interventions to improve GI function. The objective of this study was to assess transit times, pH, and pressure of the GI tract in three pigs that underwent SCI through administration of the SmartPill™ pre-injury at 2 and 6 weeks post-injury. We hypothesize that there will be delays in gastric emptying time (GET), colonic transit time (CTT), decreases in pressures in the colon, and increases in gastric and colonic pH at post-injury time points.

## 2. Materials and Methods

This study was conducted in accordance with applicable institutional and national research guidelines and regulations for the care and use of animals in research [46]. All experimental procedures were approved by the Institutional Animal Care and Use Committee of the University of Louisville (UofL) (approval number-20845). The UofL’s Animal Care and Use Program is fully accredited by the American Association for Accreditation of Laboratory Animal Care (AAALAC), International.

### 2.1. Animal and Surgical Procedures

Three 20–25 kg female Yucatan minipigs (Sinclair Bio-resources, Columbia, MO, USA) were pair-housed in floor pens and provided with a minimum of 10 ft2 per animal on 5–10 cm of Cellu-nest™ bedding (Shepherd Specialty Papers; Watertown, TN, USA) on top of 0.95 cm thick 1.22 × 1.22 m interlocking Rubber Gym Tiles (Rubber Flooring Inc., Mesa, AZ, USA). The animals were provided environmental enrichment consisting of toys, videos, and mirrors. The environmental conditions were maintained at 20.0–22.0 °C, 30–70% humidity, and a 12 h light/dark cycle. Pens were cleaned daily, and the animals were provided filtered tap water ad libitum.

Anesthesia was induced with Ketamine HCL (Zoofarm, Austintown, OH, USA, 5 mg/kg, i.m.), Dexmedetomidine (Dexmedesed®, KS, USA, 0.04 mg/kg, i.m.), and Glycopyrrolate (Piramal critical care®, PA, USA, 0.01 mg/kg, i.m.) mixed in the same syringe. Following the induction of anesthesia, Meloxicam (Covetrus®, UK, 0.4 mg/kg, i.v.) was administered for analgesia. Sustained release Bupivacaine SR (Nocita®, IN, USA, 2 mg/kg, s.q.) was divided into several locations adjacent to both sides of the planned incision site, providing local analgesia for up to 72 h. The pigs were endotracheally intubated and maintained with Propofol (PropoFlo®, NJ, USA, 8–20 mg/kg/h, i.v.) and Fentanyl (Fentanyl transdermal system, NJ, USA, 10–45 mcg/kg/h) for the entire procedure. Prior to surgery, an indwelling urinary catheter (8 French foley) was manually inserted and left in place post-operatively until the animal demonstrated continence. Standard intraoperative anesthetic monitoring recording heart and respiratory rate, blood pressure, end-tidal carbon dioxide, oxygen saturation, and urine output was conducted. The fluid status was monitored and maintained with Lactated Ringer’s solution and 5% Dextrose i.v. to maintain normoglycemia and continued post-operatively until the animals were capable of drinking independently. Dextrose was discontinued once animals were eating well. The temperature was measured by a rectal probe and maintained at 37.3–39.4 °C by a heating pad (Bair Hugger Model 775, 3M, Saint Paul, MN, USA).

With the animals in ventral recumbency on the operating table, the location of T10 was confirmed with a dorsoventral radiograph. A dorsal midline incision was made between T8 and T13. The spinous processes, laminae, and pedicles of T8 and T13 were exposed using electrocautery dissection. A second radiograph was acquired to confirm the appropriate spinal and vertebral level. Laminectomy was performed at the 10th thoracic vertebrae level to expose the dura and spinal cord and widened to a diameter of approximately 1.2 cm to ensure unimpeded impact. Two 3.5 × 14 mm multi-axial cervical screws were inserted into the T11 and T13 pedicles. Titanium rods were secured to the articulating arm of the impactor and subsequently to the pedicular screws and secured with locking caps. This fixed the T11–13 vertebral segments and secured the impactor in place and leveled it; the custom impactor was provided by University of British Columbia researchers [41]. Rocuronium was administered to mitigate animal movement during electrocautery dissection. A bolus of Propofol (10 mL bolus, equating to 100 mcg from initial concentration of 10 mcg/mL) was given five minutes prior to injury and the breath was held for impact. A 50 g cylindrical weight was dropped via a triggering mechanism onto exposed dura from a randomized height. An additional mass of 100 g was added immediately for five minutes. Following compression, the weight drop apparatus was removed and the incision was closed. Post-operatively, transdermal Fentanyl patches (Fentanyl transdermal system, NJ, USA, 1.5–5 mcg/kg/h, t.d.) and Meloxicam (Covetrus®, UK, 0.2–0.4 mg/kg, p.o.) were continued post-operatively and administered for 5 and 7 days, respectively. Maropitant (Zoofarm, Austintown, OH, USA, 1 mg/kg, p.o.) was administered pre-operatively and continued once daily for the duration of Fentanyl administration. The animals were monitored 24/7 and individually housed in open top crates in intensive care until cleared by UofL CMRU veterinarian(s) for their return to normal housing.

### 2.2. Food Diet for Study

The animals were fed Purina LabDietTM 5081 (Purina Inc., St. Louis, MO, USA) twice daily at 1% body weight per feeding for the duration of the experiment. Their food was withheld the morning of injury, resumed post-operatively initially with a/d wet dog food (Hill’s Inc., Topeka, KS, USA), and transitioned back to Purina LabDietTM 5081. The SmartPill™ was administered orally via a balling gun with the morning feeding. Twice daily feedings continued for the duration that the SmartPill™ dwelled in the animal.

### 2.3. Wireless Motility Capsule and Data Analysis

The WMC was administered randomly in the female Yucatan minipigs the week prior to initial injury (N = 2), 2 weeks post-injury (N = 3), and 6 weeks post-injury (N = 3). Female minipigs were used because of the ease of maintenance of the urinary bladder after the surgery. The WMC actively measured and transmitted pressure, pH, and temperature to a receiver secured to the animal. The pigs were monitored until capsule expulsion. The data were downloaded and the proprietary software MotiliGI® (version 2.5, Medtronic Inc., Minneapolis, MN, USA) was used to view the initial study. Proprietary Gastrointestinal Motility Software (GIMS®, version 3.0.0, Medtronic Inc., Minneapolis, MN, USA) refined the descriptive statistical analysis. The WMC was calibrated and the function confirmed using the MotiliGI® software before administration, and all of the WMCs were successfully retrieved.

The raw data generated by the MotiliGI^®^ software are displayed in (Appendix A). A drop in pH to 1–3 indicated presence in the stomach. GET was identified by a permanent rise in pH above 4. WMC transition to the small bowel was indicated by a gradual increase in pH to approximately 7–9. A sharp decline to approximately 6–8 pH with concurrent increases in pressure indicated WMC passage through the ileocecal valve and entry into the colon. The measurements were refined and validated by investigators using a GIMS^®^ data viewer for descriptive statistical analysis and stratified by anatomical section within that GI section and by time quartile.

### 2.4. Tributyrin Administration

Oral Tributyrin (Alfa Aesar Inc., Heysham, UK) at a dose of 1 g/kg/day BID was administered beginning the first day post-injury for 8 weeks as a liquid formulation mixed with flavoring emulsion (LorAnn Oils Inc., Lansing, MI, USA) in pigs 1 and 2. Previous pharmacological studies found Tributyrin to be protective against diarrhea and intestinal permeability by enhancing colonic tight junction gene expression in weaned piglets [47].

## 3. Results

### 3.1. Transit Times

Pre-injury transit times were observed for pigs 1 and 2, respectively: GETs of 4:06:30 and 7:21:10, small intestine transit times (SITTs) of 2:28:35 and 2:07:20, and CTTs of 19:55:55 and 17:27:30 (Figure 1). We observed times at 2 weeks post-injury in pigs 1, 2, and 3, respectively: GETs of 12:06:35, 99:07:37, 8:36:10, SITTs of 1:38:47, 2:14:00, and 2:39:25, and CTTs of 21:58:18, 26:25:03, and 18:08:05 (Figure 2). At 6 weeks post-injury, we recorded the following times for pigs 1, 2, and 3, respectively: GETs of 6:58:00, 8:13:51, and 8:41:40, SITTs of 2:58:05, 2:10:32, and 3:00:55, and CTTs of 36:29:15, 22:00:27, and 86:13:05 (Figure 3). These findings are clear in (Table 1), which shows delays in CTT, at both post-injury time points, and mild delays in GET in pigs 1 and 2, with indications that pig 3 was following the same trend.

### 3.2. pH Changes

Changes in post-injury pH were noted in the small intestine and colon. We observed an overall decrease in minimum (min) and median (med) pH in quartile one of the small intestine (Table 2), but the remaining gastric (Appendix A) and small intestine sections displayed no appreciable trends between pre-injury and post-injury. Colonic pH interestingly increased in both min and med pH across all quartiles and anatomical regions at 2 weeks post-injury, but returned to normality by 6 weeks (Table 3).

### 3.3. Pressure Changes

GIMS*^®^* software reported contraction frequencies (Con freq) and sum amplitudes (Sum amp), the latter sum of the peak pressure of each contraction. We observed an initial decrease in Con freq and Sum amp in the stomach at 2 weeks post-injury and, interestingly, a slight increase in the same parameters at 6 weeks post-injury. These observations illustrate the pathologic timeline and indicate potential for recovery (Table 4). The small intestine appears unimpeded by SCI, with the only notable trend in pressure being the decrease in duodenal Con freq and Sum amp that was obtained at both post-injury time points (Table 5). Decreases in colonic Con freq were observed for all quartiles at both post-injury time points, with a corresponding increase in Sum amp. In other words, there were less frequent but more forceful contractions (Table 6).

## 4. Discussion

To the best of our knowledge, this is the first study that has used the WMC in Yucatan minipigs in the context of SCI. The focus of this study was validation of the SmartPill™ in our SCI porcine model to help guide future experiments and compare existing pre-clinical and clinical GI motility studies. It is important to note that spinal injury at this level was relatively moderate and resulted in complete loss of hindlimb function. CTTs are generally longer in non-injured pigs when compared to humans, but similarly variable [37,48]. T10–T11 level SCI in pigs 1 and 2 (Table 1) was associated with disturbances of GET and CTTs, as hypothesized. Pre-injury data for pig 3 was unobtainable. It is well established that SCI at a T4–5 level is associated with significant GI disturbance including pathologic gastroparesis [5,16]. The majority of pre-clinical GI motility SCI studies have been performed in rodents, specifically rats, and depict a prolongation of both the gastric and colonic times after injury [16,49,50]. While the mechanism behind gastric and colonic delays remains unclear, previous findings indicate that autonomic dysfunction via alterations in vagal nerve signaling integrity post-injury plays a significant role [5,15,16,51,52,53,54]. This is further confirmed by Schneider et al., who also showed a prolongation of GET that was greater than CTTs or SITTs [37].

We noted overall decreases in colonic contraction frequencies across all quartiles but, interestingly, an increase in Sum amp. This indicates that the colonic and gastric motor complexes are in an aberrant state, potentially contributing to delays in transit times and leading to a constipation also likely exacerbated by opioid use for analgesia after SCI [55]. Another trend was the increase in colonic pH in all quartiles at 2 weeks post-injury and a return to baseline at 6 weeks. This indicates a disturbance in the microenvironment in the acute time frame, with potential for recovery. In non-injured conditions, the colonic pH in both male Landrace and our pre-injury female Yucatan minipigs was comparable to that of humans and generally ranged from 5 to 8 [35]. GI dysbiosis has recently gained attention as a source of GI disturbances. The gut microbiome’s normal function significantly impacts cognition, digestion and absorption, and overall wellness. In the context of SCI, the gut microbiome is altered and affects the clinical prognosis [19,56,57]. Thus, pharmacological influence of the biome is of interest. We administered oral Tributyrin, a triglyceride that reportedly encourages natural gut flora, to assess its impact on motility and the colonic environment. We would expect that pigs receiving Tributyrin would display improved colonic pH and CTTs. However, the effect of Tributyrin is unclear in our study, likely due to the small sample size, and further studies are required.

We observed a significant decrease inCon freq and Sum amp in the antrum at 2 weeks post-injury and a slight improvement in these parameters at 6 weeks post-injury. These findings indicate that there is significant deterioration or a “shock” period of autonomic function in the acute setting that partially abates by 6 weeks. This change over time suggests potential for improvement. Maximum pressures in both non-injured humans and large canines have been reported to range from 100 to 500 mmHg in the fed state [37,58]. Male Landrace pigs’ gastric pressures were reported to range from 100 to 350 mmHg in the fed state [35]. Another study conducted by Raunch et al. in Pietrain farm pigs reported pressures between 4 and 20 mmHg [59]. It is unclear why these gastric pressures are substantially lower; it is likely a result of the experimental conditions. Beagle dog maximum gastric pressures have also been reported to range from approximately 200 to 800 mmHg [48].

Closer examination of the 2 weeks post-injury graph for pig 2 (Figure 2) shows the WMC dwelling in the stomach for an extended time and multiple unsuccessful attempts to expel the capsule over four days. This event could have stemmed from repetitious feedings, which rhythmically close the pylorus under normal physiologic conditions. The unique U shape of the pig stomach can impede passage of larger, more solid objects by the pyloric sphincter [35,60]. Having observed decreases in antrum pressures after injury, this could further exacerbate the passage of the WMC and solid foods into the duodenum and contribute to pathology development. Pig 3 also displayed a significant delay in the colon compared to other animals. This is likely an extreme effect of injury on colonic motility; however, it could also stem from the unique spiral shape of the porcine colon. On the other hand, the literature utilizing porcine GI models indicates that this feature does not appreciably affect motility [61]. We considered the effect of fentanyl administration on GI motility; however, the published pharmacokinetic and pharmacodynamic parameters in humans indicate that relatively large doses of fentanyl (100 g/h) administered transdermally and continuously over a week were present in negligible amounts after 6 days. Given that the animals had fentanyl removed 5 days after injury, we concluded that fentanyl would be cleared from the system by 2 weeks post-injury and was not responsible for the changes from week 2 to week 6 of administration [62].

The small intestine was largely unaffected when comparing pre- and post-injury pressures, except for decreased contraction frequencies and maximum pressures in the duodenum and quartile 1. The Yucatan minipigs in this study and a previous Beagle canine study displayed shorter SITTs of 1–3 h when compared to humans and a marginally shorter relative to male Landrace pigs in the fed condition [35,37]. It is suggested that the shorter intestinal transit times can be correlated to different dimensions of intestine. In dogs, the small intestine is shorter compared to humans and pigs, but this does not appear to affect transit [31,60]. These data show that small intestine motility is essentially unaffected after SCI, and this agreed with previous observations in humans, rodents, and other species studies to date [35,37,48,58,59]. We observed transient increases in small intestine pH in all quartiles and anatomical locations supposed by GIMS*^®^* software at 2 weeks post-injury that returned to pre-injury conditions by 6 weeks. The small intestine pH in male Landrace pigs was marginally higher and comparable to our limited pre-injury Yucatan data [35]. Two recently published studies involving a WMC in large canines and beagles revealed a baseline colon and small intestinal pHs that were comparable to those of humans [31,58]. Changes in small intestinal absorptive and microenvironment physiology likely play a role in pathogenesis after SCI. This change is important to consider clinically as these changes influence nutrimental status and drug absorption, features that can directly affect prognosis of SCI patients [5,12,13]. Minimum gastric pH was similar across all dog and pig species and relative to humans, ranging from 0.1 to 1, with the exception of the male Gottingen minipig study by Suenderhauf et al., which reported min gastric pHs ranging from 1.2 to 6 [30,31,59,60,63].

Williams et al. administered SmartPill™ to 20 patients in a chronic cervical and thoracic SCI setting with a mean injury duration of 15 *±* 11 years [39]. Patients were fed after a 12-h fast and were asked to swallow a smart pill to measure the GET and the CTT. They reported a prolonged GET of 10.6 *±* 7.2 h in patients with SCIs versus 3.5 *±* 1.0 h in control subjects. Similarly, CTT was 52.3 *±* 42.9 h versus 14.2 *±* 7.6 h in controls. Additionally, they reported no significant change in post-injury min gastric pH after injury. This study concluded that the SmartPill™ was safe and can be used to demonstrate gastrointestinal motility delays in patients with both cervical and thoracic SCI [39]. Delays in GET and CTT and general trends in GI motility after SCI from previous studies agree with our observations in the Yucatan minipig model post-injury. Further comparisons are limited by the paucity of published data on pressure and pH parameters beyond MotiliGI*^®^*.

Although this study was a preliminary study to evaluate the Smartpill^TM^ function in the minipig model of SCI, a major limitation of this study was the sample size. The study was conducted to evaluate and test the feasibility of the Smartpill^TM^ in a pre-clinical model of SCI. Moreover, in this study, we did not consider the representability of the minipig gut with human. This could give us more insight into the gastrointestinal transit time information. Another limitation of the study is that the anatomical structure of the minipig spinal cord was not compared in this study, while some pieces of information are available only for domestic pigs. The nerves that supply to the gastrointestinal tract from the spinal cord were not very well known in Yucatan minipigs. Additionally, the administration of tributyrin can modulate the transit time and pH. Further study will elucidate the effect of tributyrin on the microbiome.

## 5. Conclusions

This is the first pre-clinical study that has implemented the SmartPill™ in the context of SCI using a large animal model. Our trends post-injury are consistent with the limited data that exist, both in pre-clinical and human studies. There are delayed GETs, CTTs, Con freq, and Sum amp in both the antrum and colon and a transient increase in colonic pH 2 weeks after injury. The potential impact of Tributyrin remains unclear, and further studies are necessary to elucidate its influence post-injury. We acknowledge the limited power of this pilot study, and additional studies are required to establish larger trends. Further investigations are warranted to establish these trends in a larger cohort, at which point GI microenvironmental changes can be effectively modulated to improve gut function and overall morbidity.

## Figures and Tables

**Figure 1 biomedicines-11-01660-f001:**
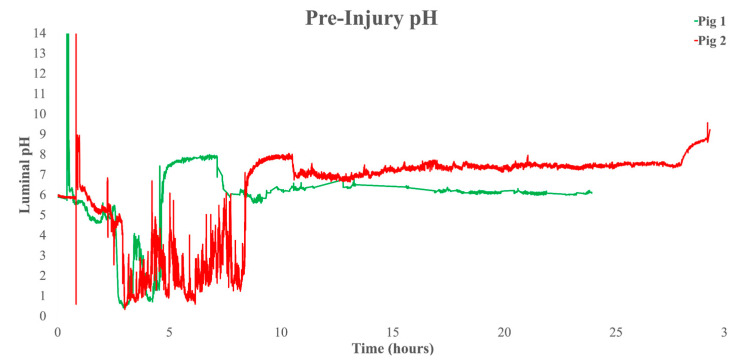
Whole gut pH recorded by the SmartPill™ at pre-injured condition. The green line depicts pig 1 and the red line pig 2. Graphs include raw data recorded by the SmartPill™. A drop in mean pH to between 1 and 3 indicates that the wireless motility capsule (WMC) has been ingested and resides in the stomach. A permanent rise in mean pH to above 4 to between approximately 7 and 9, with a concomitant increase in pressure, indicates passage through the pyloric sphincter GET and into the small intestine. A subsequent decrease in pH to approximately 6 to 8, with a congruent increase in pressure, indicates passage through the ileocecal valve SITT into the colon, indicating colon arrival time (CAT) and the start of CTT. A permanent and continuous rise above pH 7 indicates that the WMC exited the body, completing whole gut transit time (WGTT).

**Figure 2 biomedicines-11-01660-f002:**
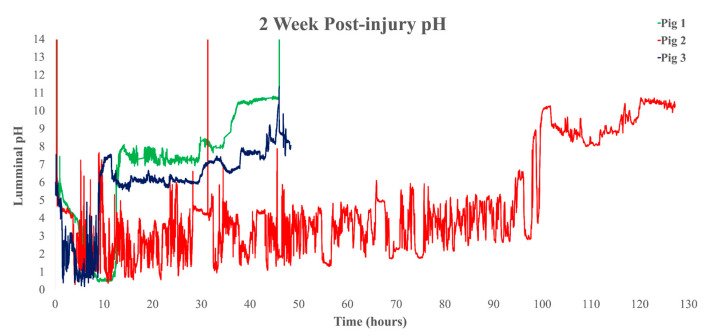
Whole gut pH recorded by the SmartPill™ at week 2 post-injured condition. The green line is pig 1, the red line is pig 2, and the dark blue is pig 3. Graphs include raw data recorded by the SmartPill™, with previously described graph analysis.

**Figure 3 biomedicines-11-01660-f003:**
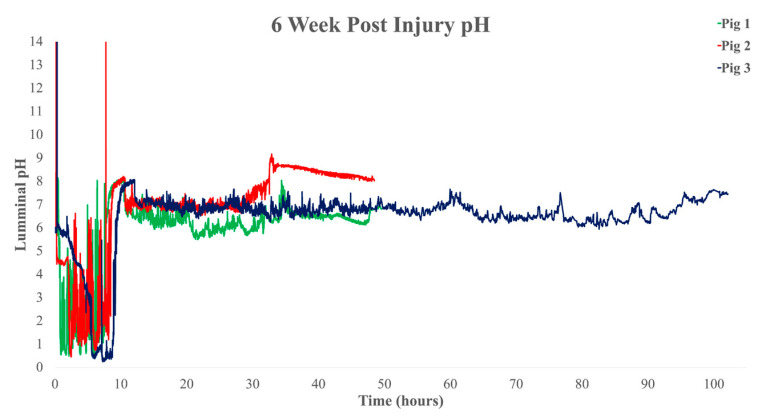
Six weeks post-injury whole gut pH recorded by the SmartPill™. The green line is pig 1, the red line is pig 2, and the dark blue is pig 3. Graphs include raw data recorded by the SmartPill™, with previously described graph analysis.

**Table 1 biomedicines-11-01660-t001:** Summary of transit times recorded in hours by the SmartPill® at each time point for each animal. ^1^ Colon Transit Time, ^2^ Gastric Emptying Time, ^3^ Small Intestine Transit Time, ^4^ Colon arrival time, ^5^ Whole Gut Transit Time.

Pig	Time Frame Relative to Injury	^1^ GET	^2^ SITT	^3^ CTT	^4^ CAT	^5^ WGTT
1	Pre-injury	4:06:30	2:28:35	19:55:55	6:35:05	26:29:00
	2 weeks post	12:06:35	1:38:47	21:58:18	13:45:22	35:43:40
	6 weeks post	6:58:00	2:58:05	36:29:15	11:34:15	48:03:30
2	Pre-injury	7:21:10	2:07:20	17:27:30	9:28:30	26:56:00
	2 weeks post	99:09:57	2:09:50	26:26:53	101:21:47	127:48:40
	6 weeks post	8:10:35	2:10:30	22:12:55	10:21:05	32:43:00
3	2 weeks post	8:36:10	2:39:25	18:08:05	11:15:35	29:23:40
	6 weeks post	8:41:40	3:00:55	86:13:05	11:42:35	97:55:40

**Table 2 biomedicines-11-01660-t002:** Small intestine min and med pH recordings stratified by time quartiles and specific anatomical colonic regions of interest for each animal at each time point after WMC administration.

		Duodenum		Ileum		Quartile 1		Quartile 2		Quartile 3		Quartile 4	
Pig	Time	Min	Med	Min	Med	Min	Med	Min	Med	Min	Med	Min	Med
1	Pre-injury	1.95	7.58	7.73	7.9	1.95	7.46	7.6	7.72	7.67	7.82	7.73	7.9
	2 weeks post	1.37	7.47	6.9	7.9	1.37	6.14	6.05	7.25	7.65	7.82	7.75	8.01
	6 weeks post	1.43	6.71	7.9	7.99	1.43	6.56	7.29	7.71	7.82	7.96	7.9	7.97
2	Pre-injury	2.12	7.84	7.74	7.88	2.1	7.08	7.42	7.69	7.74	7.86	7.74	7.88
	2 weeks post	7.86	9.99	10.12	10.24	7.86	9.42	10.01	10.12	10.12	10.18	10.22	10.26
	6 weeks post	2.22	7.74	7.84	8.07	2.22	7.42	7.73	7.86	7.84	7.99	7.95	8.09
3	2 weeks post	1.29	6.52	7.29	7.46	1.29	6.35	6.42	6.84	7.12	7.35	7.39	7.48
	6 weeks post	2.26	6.67	7.82	7.97	2.26	6.33	7.01	7.63	7.71	7.88	7.88	7.99

**Table 3 biomedicines-11-01660-t003:** Colonic min and med pH recordings, further stratified by quartile of colon and specific anatomical location for each animal at each time point after WMC was administered.

		Caecum		Sigmoid		Quartile 1		Quartile 2		Quartile 3		Quartile 4	
Pig	Time	Min	Med	Min	Med	Min	Med	Min	Med	Min	Med	Min	Med
1	Pre-injury	5.9	6.69	6.07	6.23	5.61	6.14	6.13	6.39	6.01	6.13	5.95	6.11
	2 weeks post	7.43	7.63	8.2	8.28	6.93	7.56	6.94	7.29	6.95	7.29	7.67	8.12
	6 weeks post	6.48	6.77	6.15	6.21	5.95	6.48	5.5	5.95	5.67	6.48	6.15	6.49
2	Pre-injury	6.76	7.08	7.35	7.5	6.67	6.99	7.11	7.39	7.14	7.43	7.31	7.5
	2 weeks post	9.05	9.22	10.03	10.24	8.27	8.89	8.03	8.56	8.65	9.61	9.92	10.41
	6 weeks post	6.59	6.97	7.22	8.01	6.59	7.01	6.63	6.92	6.56	6.96	6.63	7.41
3	2 weeks post	6.05	6.27	5.92	6.01	5.52	5.97	5.95	6.16	5.71	6.09	5.92	6.05
	6 weeks post	6.88	7.24	6.99	7.18	6.25	6.9	6.37	6.78	6.22	6.56	5.95	6.47

**Table 4 biomedicines-11-01660-t004:** Gastric pressure recordings stratified by time quartiles and anatomical regions of interest for each animal at each time point after WMC administration. ^1^ Contraction Frequency, ^2^ Summation of the Amplitudes.

		Antrum		Quartile 1		Quartile 2		Quartile 3		Quartile 4	
Pig	Time	^1^ Con Freq	^2^ Sum Amp	Con Freq	Sum Amp	Con Freq	Sum Amp	Con Freq	Sum Amp	Con Freq	Sum Amp
1	Pre-injury	12.6	13,541.92	6.96	6224.81	5.17	3701.71	5.85	4125.89	12.25	13,542.63
	2 weeks post	2.64	7204.76	3.14	9589.59	4.29	12,143.88	4.55	14,139.2	2.82	13,267.17
	6 weeks post	7.71	12,609.13	3.17	5185.17	2.77	7703.32	4.05	16,758.34	7.37	20,221.82
2	Pre-injury	6.1	8624.08	2.84	4896.96	3.21	10,021.52	2.29	5614.76	4.95	13,141.26
	2 weeks post	1.47	5920.24	2.42	79,795.96	2.26	87,694.51	1.89	64,920.47	1.82	53,130.43
	6 weeks post	2.88	5928.46	1.46	2362.98	3.32	16,949.41	4.47	12,145.5	3.85	11,257.91
3	2 weeks post	2.75	11,208.56	5.1	10,271.68	3.27	7069.23	3.82	9694.13	3.39	23,596.8
	6 weeks post	4.03	7358.97	3.32	7400.25	2.29	5747.71	3.67	9257.94	3.37	11,169.59

**Table 5 biomedicines-11-01660-t005:** Small intestine pressure recordings further stratified by time quartiles and specific regions for each animal at each time point after WMC administration.

		Duodenum		Ileum		Quartile 1		Quartile 2		Quartile 3		Quartile 4	
Pig	Time	Con Freq	Sum Amp	Con Freq	Sum Amp	Con Freq	Sum Amp	Con Freq	Sum Amp	Con Freq	Sum Amp	Con Freq	Sum Amp
1	Pre-injury	5.49	4806.65	1.77	1073.03	7.59	3948.92	2.29	1286.37	2.57	823.44	1.49	714.42
	2 weeks post	5.08	5123.13	5.35	4140.09	3.34	2105.76	7.03	2459.54	7.25	1614.31	3.66	1736.16
	6 weeks post	1.44	1839.88	3.9	3740.14	0.81	941.3	4.54	3386.94	1.81	1568.18	4.87	3440.62
2	Pre-injury	2.2	2456.44	0.59	844.1	3.5	2132.12	0.96	380.9	0.84	688.16	0.42	241.96
	2 weeks post	0.89	920.92	0.41	421.82	1.24	695.06	0.53	266.8	0.19	123.72	0.68	325.22
	6 weeks post	1.1	1835.98	0.47	456	1.35	1566.22	0.73	293.76	0.49	202.56	0.44	275.52
3	2 weeks post	1.67	1821.23	0.57	659.86	1.51	1166.52	1.97	1140.22	0.92	548.96	0.51	438.47
	6 weeks post	2.2	3139.08	1.15	1145.38	2.23	2655.14	1.27	881.84	0.96	659.2	1.04	814.2

**Table 6 biomedicines-11-01660-t006:** Colonic pressure recordings further stratified by time quartiles and anatomical region for each animal at each time point after WMC administration.

		Quartile 1		Quartile 2		Quartile 3		Quartile 4	
Pig	Time	Con Freq	Sum Amp	Con Freq	Sum Amp	Con Freq	Sum Amp	Con Freq	Sum Amp
1	Pre-injury	1.59	950.81	1.76	34.86	1.84	1721.61	1.44	1052.29
	2 weeks post	2.26	1820.64	3.09	3776.22	2.59	1910.86	1.34	614.88
	6 weeks post	0.08	1656.46	0.04	660.02	0.16	2522.4	0.14	2912.1
2	pre-injury	0.56	356.5	0.49	614.1	0.59	499.56	1.67	1090.66
	2 weeks post	0.08	911.7	0.05	508.76	0.31	3543.36	0.4	5234.8
	6 weeks post	0.06	765.6	0.08	842.4	0.14	1218.24	0.42	4306.1
3	2 weeks post	0.51	1178.29	1.16	1563.24	2.22	2458.04	1.6	2547.76
	6 weeks post	0.17	5298.34	0.05	1546.96	0.03	1050.62	0.09	3143.62

## Data Availability

All data of this study are available from author upon request.

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
