# Peer review of "SmartPill™ Administration to Assess Gastrointestinal Function after Spinal Cord Injury in a Porcine Model—A Preliminary Study"

_biomedicines, 2023, doi:10.3390/biomedicines11061660_

Round 1

Reviewer 1 Report

In the present experimental study Knibbe et al used Smartpill to evaluate luminal pH, temperature and pressure in pigs with spinal cord injury (SCI). They found delayed gastric emptying time and colon transit time and increased duodenal pressures after SCI. Main comments:

1) Page 8 line 235: why data were unavailable for pig 3?

2) Do not use °F, use °C

3) Could improvement of GET at 6w, compared to 2w, be justified by the end of the effect of opioids used during anaesthesia, rather than by adaptation after surgery? Please discuss.

4) Limitations of the study should be better discussed. For example, only 3 pigs were analyzed. Therefore this could be considered as a preliminary study, and this should be acknowledged even in the title.

Author Response

Dear Reviewers,

we would like to sincerely thank the reviewer for all the constructive and invaluable comments, and we really appreciate the reviewer suggestions to improve our manuscript.

Reviewer 2 Report

This is an interesting study important from practical point of view. However, the study contains some shortness’s which need to be answered. Also, an issue of potential COI needs to be explained.

Line 68 - Please provide Ethic Committee Approval number

Line 73 – I think that the main limitation of this study is small number of experimental animals used. The authors should carefully explain why only three animals were studied. What did the selection of the number of animals depend on? Furthermore, the issue how to establish the minimal number of animals to test to get statistically significant data is really disputable. Although, there are some directions provided by Universities or Journals (https://www.nature.com/articles/laban0508-193a) most of researchers agree that depending on kind of experiment the minimal number of experimental animals should be 5 or 6 (I personally agree with this). Thus, n=3 is just too small. Another question why females only?

Line 82 – please change IM into i.m., IV into i.v. etc.

Line 95 – please provide temperature values in oC

Line 97 – Please specify what is T10? (tenth thoracic vertebrae?)

Line 199 – Caecum is not a part of the small intestine as declared in title. What is “Sigmoid”???

Line 152 and more – the authors should ensure that they use term “expression” in relation to genes only.

Line 170, 188 and more – There is only one small intestine (divided into three parts – duodenum, jejunum and ileum)

Author Response

Dear Reviewer,

we would like to sincerely thank the reviewer for all the constructive and invaluable comments, and we really appreciate the reviewer suggestions to improve our manuscript.

Round 2

Reviewer 1 Report

Answers were fine